# Fate of Pomeranchuk effect in ultrahigh magnetic fields

Naofumi Matsuyama[1] ✉, So Yokomori [2,3,6], Toshihiro Nomura[4], Yuto Ishii [1], Hiroaki Hayashi[1], Hajime Ishikawa[1], Kazuki Matsui[1], Hatsumi Mori [1], Koichi Kindo[1], Yasuhiro H. Matsuda [1] & Shusaku Imajo [1,5] ✉

The Pomeranchuk effect is a counterintuitive phenomenon where liquid helium-3 ($^3$He) solidifies under specific pressures, not when cooled, but when heated. This behaviour originates from the magnetic entropy of nuclear spins, suggesting a magnetic field should influence it. However, its detailed response to magnetic fields remains elusive due to the small nuclear magneton of $^3$He and lack of analogous fermion systems. Here, we show that an electron system also exhibit the Pomeranchuk effect, where the Fermi liquid state solidifies in a high magnetic field, unlike conventional electron systems where a field melts an electron solid into a metal. Remarkably, the electron system displays a reentrant liquid state in ultrahigh fields. These responses are explained by changes in magnetic entropy and magnetisation, extending the underlying physics to $^3$He. Our findings clarify magnetic-field impact on the Pomeranchuk effect and open avenues for magnetic control of chemical interactions.

In 1950, Pomeranchuk proposed that extremely low-temperature liquid $^3$He could be further cooled upon solidification by applying pressure[1]. The nucleus of $^3$He consists of two protons and one neutron, making $^3$He a fermion. As a result, liquid $^3$He behaves as a Fermi liquid, exhibiting entropy proportional to temperature, $S = \gamma T$, where $S$ and $\gamma$ represent the entropy and Sommerfeld coefficient, respectively. When solidified under high pressure, localised $^3$He atoms with nuclear spins $I = 1/2$ form a paramagnet whose entropy remains nearly constant at $S = R\ln 2$ above 20 mK (Fig. 1a). A comparison of the entropies between the solid and liquid states reveals that the slope of the melting curve $(dP/dT)_\text{melt}$ becomes negative below 0.32 K, as illustrated in Fig. 1b. Consequently, at a certain pressure (e.g., 30 bar), increasing the temperature results in entropically driven phase transition to a solid state, in stark contrast to the thermally driven melting observed in typical ordered systems. This counterintuitive phenomenon is known as the Pomeranchuk effect[2].

The Pomeranchuk effect is expected to be influenced by magnetic fields due to the magnetic degrees of freedom associated with nuclear spins. This suggests that magnetic-field-induced melting or

solidification could occur in $^3$He. Although previous studies investigated the melting curve of $^3$He in magnetic fields for its use as a low-temperature thermometry standard[3,4], the observed changes were minimal because the applied fields were insufficient to polarise the nuclear spins. Hence, the detailed magnetic-field response of the Pomeranchuk effect remains elusive.

Given the universality of physics, analogous systems exhibiting the Pomeranchuk effect are expected to exist and may serve as useful platforms for exploring their magnetic-field response. In particular, electron systems provide significant advantages over $^3$He for studying magnetic field effects, as the Bohr magneton of an electron is 1840 times larger than the nuclear magneton of $^3$He, making electron systems far more sensitive to magnetic fields. For electrons, which are also fermions, a liquid-solid transition corresponds to a metal-insulator transition (MIT), where they are itinerant in the metallic state and localised in the insulating state. When MITs are driven by strong correlations originating from Coulomb repulsion, these correlations manifest as kinetic exchange interactions in the insulating phase, releasing magnetic entropy at high temperatures. As a result,

[1]Institute for Solid State Physics, University of Tokyo, Chiba, Japan. [2]College of Science, Rikkyo University, Tokyo, Japan. [3]Research Center for Smart Molecules, Rikkyo University, Tokyo, Japan. [4]Department of Physics, Faculty of Science, Shizuoka University, Shizuoka, Japan. [5]Department of Advanced Materials Science, University of Tokyo, Chiba, Japan. [6]Present address: Graduate School of Science and Engineering, Ibaraki University, Ibaraki, Japan. ✉e-mail: naofumi.matsuyama.1009@gmail.com; imajo@edu.k.u-tokyo.ac.jp

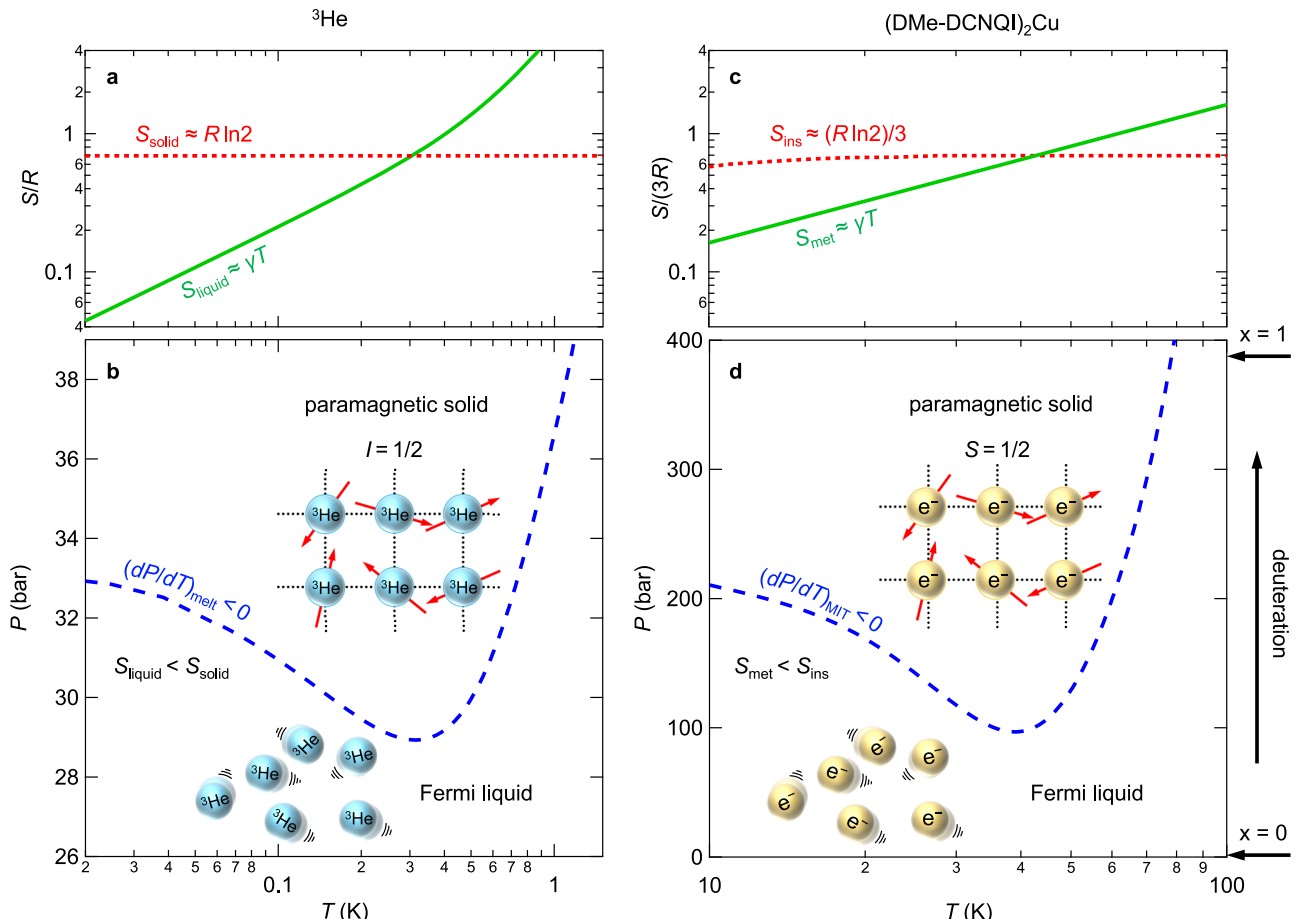

**Fig. 1 | Entropy and liquid-solid phase diagrams of $^3$He and (DMe-DCNQI)$_2$Cu at zero field.** Temperature dependence of entropy (**a**) and melting curve on temperature-pressure phase diagram (**b**) of $^3$He. Below 0.32 K, the entropy of solid exceeds that of liquid due to magnetic entropy of non-interacting nuclear spins, resulting in the negative slope of the melting curve $(dP/dT)_{melt} < 0$, known as the Pomeranchuk effect, as shown in (**b**). Temperature dependence of entropy (**c**) and melting curve on temperature-pressure phase diagram (**d**) of [(DMe-DCNQI)$_{1-x}$($d_8$-DMe-DCNQI)$_x$]$_2$Cu. The vertical axis variable, pressure, can also be tuned by chemical pressure induced by deuterium substitution, as shown on the right vertical axis of (**d**).

conventional correlated systems do not exhibit the Pomeranchuk effect. Only a limited number of electronic systems exhibiting the Pomeranchuk-like effect have been reported[5–12], yet their circumstances differ significantly from those of $^3$He. For instance, in the Mott transition observed in an organic material[7], the strong magnetic interactions in the insulating phase mask the intrinsic magnetic-field effects. To further deepen our understanding of the innate characteristics of the Pomeranchuk effect, it has been desirable to explore another condensed-matter system that faithfully reproduces its original manifestation in $^3$He.

Here, we revisit the electronic states of the coordination polymer (DMe-DCNQI)$_2$Cu, where DMe-DCNQI represents 2,5-dimethyl-N,N'-dicyanoquinonediimine. One-dimensional DMe-DCNQI columns coordinated by Cu ions form a tetragonal lattice of space group $I4_1/a$, showing metallic behaviour down to the lowest temperatures arising from a hybridised electron band composed of DCNQI π-electrons and Cu d-electrons[13–19]. External pressure or chemical substitution can shift the ground state into a Cu$^{2+}$-originated paramagnetic insulator accompanied by a three-fold superlattice formation, with an intermediate range where a reentrant metal-insulator-metal transition is observed (Fig. 1d)[13–15]. The reentrant metallic ground state has been identified to share the same electronic state as the metallic state observed in other temperature and pressure regions[20]. While an earlier study has investigated the entropic relation between metallic and insulating states, which explains the

emergence of the reentrant metallic phase, its link to the Pomeranchuk effect remains unexplored[19]. Given the paramagnetic insulator state of (DMe-DCNQI)$_2$Cu with tiny magnetic interactions, analogous to solid $^3$He, the reentrant metallic state can be viewed as one manifestation of the Pomeranchuk effect. Indeed, as shown in Fig. 1c, d the temperature dependence of the entropy and phase diagram of (DMe-DCNQI)$_2$Cu[19] closely resembles those of $^3$He (Fig. 1a, b). To explore this connection, we examined (DMe-DCNQI)$_2$Cu from the perspective of the electronic Pomeranchuk effect and investigated its response to magnetic fields. Chemical pressure, the counterpart of physical pressure in $^3$He, was introduced by mixing deuterated DMe-DCNQI ($d_8$-DMe-DCNQI)[14]. The deuteration of DMe-DCNQI acts as a positive chemical pressure, as shown in Fig. 1d. Mapping the field-temperature ($H$-$T$) phase diagram of [(DMe-DCNQI)$_{1-x}$($d_8$-DMe-DCNQI)$_x$]$_2$Cu alloy, we observed a field-induced solidification of electrons, suggesting the Pomeranchuk effect is suppressed under magnetic fields. In the ultrahigh-field limit, the system exhibits a reentrant metallic state, hinting at a possible emergence of a novel quantum liquid state of $^3$He. Our findings clarify the behaviour of Pomeranchuk electrons under magnetic fields, revealing their fate in extreme conditions.

## Results
To begin with, we characterised the fundamental transport properties of the [(DMe-DCNQI)$_{1-x}$($d_8$-DMe-DCNQI)$_x$]$_2$Cu alloy ($x = 0.3$, 0.5, and 1).

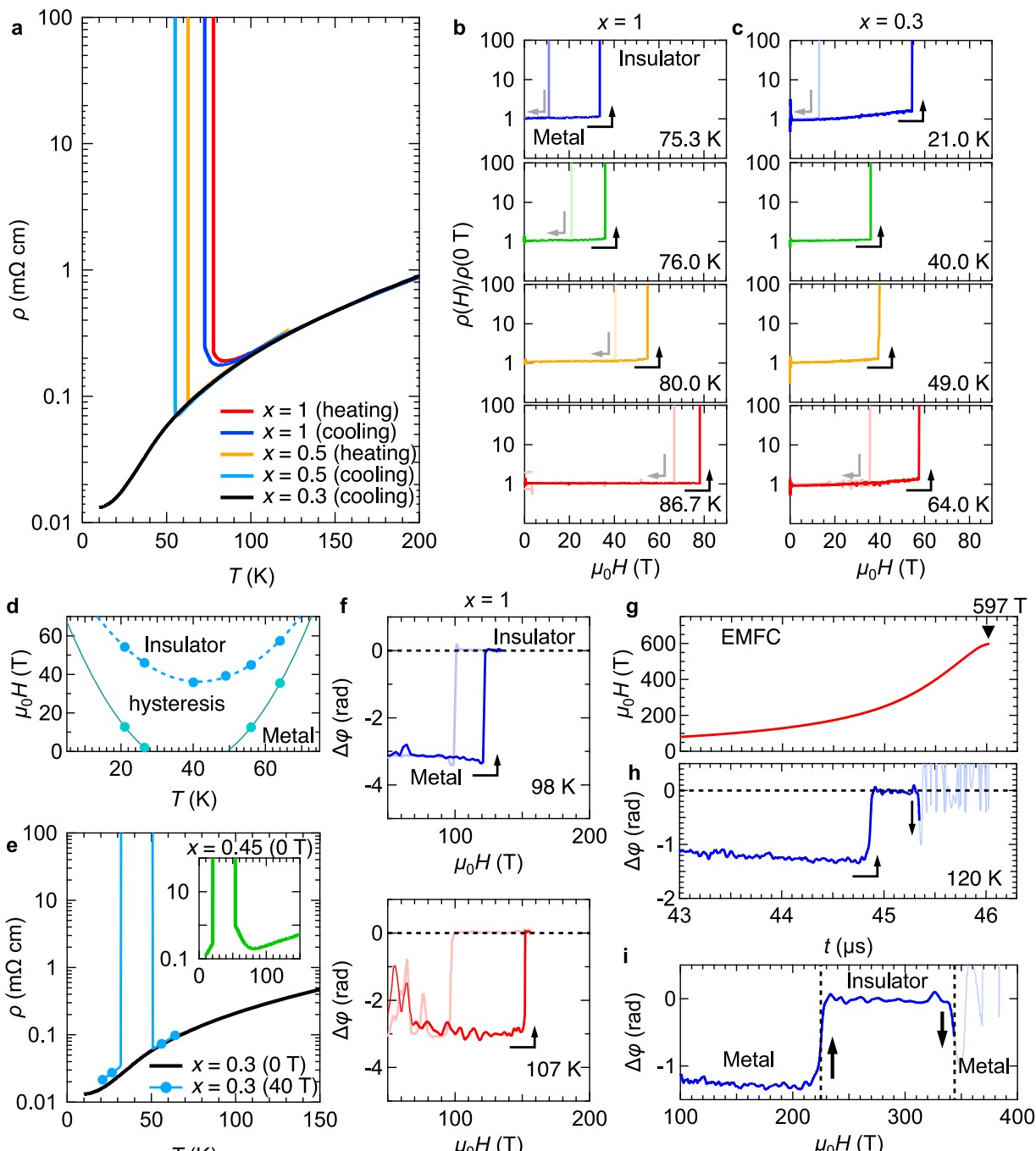

**Fig. 2 | Electrical transport properties. a** Zero-field electrical resistivity of [(DMe-DCNQI)$_{1-x}$($d_8$-DMe-DCNQI)$_x$]$_2$Cu ($x$ = 0.3, 0.5, and 1) as a function of temperature. Electrical transport measurements in pulsed fields up to 88 T at various temperatures for the $x$ = 1 (**b**) and $x$ = 0.3 (**c**) salts. Dark-coloured curves correspond to data obtained during field-ascending sweeps, while light-coloured curves correspond to data obtained during field-descending sweeps. Arrows indicate directions of field sweeps. In some cases, due to large hysteresis, the insulating state persists down to zero field. **d** Phase boundary between metallic and insulating states of $x$ = 0.3, determined from (**c**). **e** Temperature-dependent electrical resistivity of $x$ = 0.3 at 0 T (black) and 40 T (blue). Inset shows data for $x$ = 0.45 at 0 T. **f** Magnetic field dependence of the relative phase change $\Delta\varphi$ of RF waves in impedance measurements using STCs. Time evolution of a pulsed magnetic field generated by an EMFC system (**g**) and $\Delta\varphi$ of the $x$ = 1 salt in the pulsed field (**h**). **i** Magnetic field dependence of $\Delta\varphi$ obtained by the EMFC system shown in (**g**, **h**).

Figure 2a displays the zero-field resistance of [(DMe-DCNQI)$_{1-x}$($d_8$-DMe-DCNQI)$_x$]$_2$Cu as a function of temperature at ambient pressure. For the fully deuterated salt ($x$ = 1) and $x$ = 0.5 salt, sharp resistance jumps at $T$ ~ 80 K and 60 K, respectively. It is accompanied by hysteresis, signifying a first-order MIT. To simplify the analysis of the external field dependence of the MIT, the transition point is defined during the cooling process unless otherwise stated. In contrast, the $x$ = 0.3 salt

exhibits monotonic metallic behaviour down to 10 K. These results, consistent with the prior study[14], demonstrate that the alloy systems enable access to the finite-pressure conditions under ambient pressure, which provides a basis for exploring their response under magnetic fields.

Next, we explored the effect of magnetic fields on the DCNQI family. Figure 2b presents the results of electrical resistivity ($\rho$)

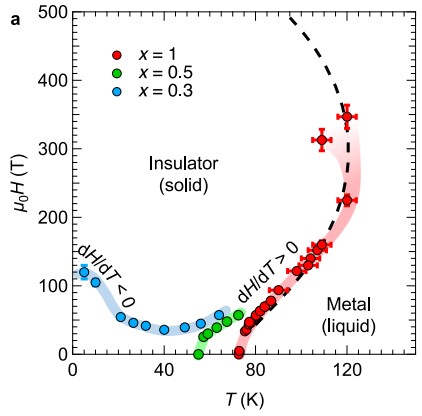

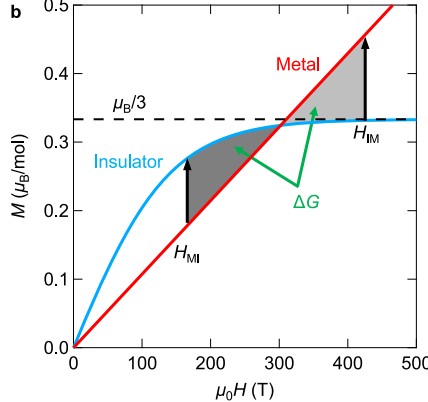

**Fig. 3 | Phase diagram of Pomeranchuk electrons. a** Magnetic-temperature metal-insulator phase diagram of $[(DMe\text{-}DCNQI)_{1-x}(d_8\text{-}DMe\text{-}DCNQI)_x]_2Cu$ (blue: $x = 0.3$, green: $x = 0.5$, and red: $x = 1$). Bars attached to data points represent uncertainties in temperature and magnetic field. Temperature uncertainty arises from heating during a field generation (see Supplementary Material), whereas magnetic field uncertainty reflects the electromagnetic noise in STC experiments (see Supplementary Material for raw data) and possible deviation from the field centre due to sample length in EMFC experiments[35]. Dashed curve is a phase boundary phenomenologically estimated for the $x = 1$ salt. Thick curves superimposed on the data points serve as visual guides. **b** Simulation of magnetisation curves in metallic (red) and insulating (blue) states. Dashed line denotes saturation of the insulating state at $\mu_B/3$. Black arrows indicate the magnetisation jump associated with fields of metal-insulator and insulator-metal phase transitions, $H_{MI}$ and $H_{IM}$. The areas of the two grey regions, representing energy gains $\Delta G$ due to magnetic fields, are equal.

measurements for the $x = 1$ salt in pulsed fields generated by non-destructive magnets. The field-induced MITs are first-order phase transitions accompanied by hysteresis. The wide hysteresis indicates that nucleation associated with the phase transition requires overcoming an energy barrier, whereas the sharpness of the transition implies that domain growth develops rapidly. The transition field $H_{MIT}$ shifts to higher fields as the temperature increases. A similar behaviour is observed in the $x = 0.5$ salt as well (see Supplementary Materials). Meanwhile, as shown in Fig. 2c, the $x = 0.3$ salt, which remains metallic across all temperatures at 0 T, underwent a field-induced MIT. The phase boundary, shown in Fig. 2d, exhibits a minimum in the transition field $H_{MIT}$ around 40 K, which corresponds to the temperature at which the entropies of the insulating and metallic states become equal (see Fig. 1c). Notably, near this temperature, the insulator-to-metal transition (IMT) during the field-descending process is absent due to the pronounced hysteresis. As illustrated in Fig. 2e, the $x = 0.3$ salt is metallic throughout the temperature range at 0 T. However, under a high magnetic field of 40 T, it exhibits reentrant MIM transitions, similar to the behaviour observed for the $x = 0.45$ salt at 0 T (inset of Fig. 2e).

To probe the field-induced MITs beyond the range accessible by non-destructive pulsed magnets, we performed impedance measurements using a single-turn coil (STC) system, extending the field range up to approximately 260 T. This technique detects resistance changes through variations in the reflection wave amplitude and phase rotation of injected RF waves[20] (see Supplementary Materials, SM). Figure 2f illustrates the relative phase change $\Delta\varphi$ of RF waves as a function of the magnetic field at 98 K and 107 K, measured in pulsed fields generated by the STC system. Consistent with the electrical resistivity results in Fig. 2b, c, field-induced MITs were clearly observed as amplitude change and phase rotation of RF waves in the impedance measurements.

To push the exploration into the ultrahigh-field regime, we conducted impedance measurements on the $x = 1$ salt in high fields of up to 600 T using an electromagnetic flux compression (EMFC) system[21] (see Methods and SM). Figures 2g, h illustrate the time dependence of the generated magnetic field and $\Delta\varphi$ during the pulsed field, respectively. When the field reached 220 T ($t = 44.8$ μs), a positive phase rotation was observed, corresponding to the field-induced MIT detected in the impedance measurements using STCs. As the applied field increased further, reaching approximately 320 T

(45.4 μs), the phase began to revert toward its low-field values, signalling the onset of a field-induced insulator-metal transition (IMT). Shortly afterward, the RF signal collapsed (highlighted in light colour in Fig. 2h, i), notably before the coil explosion at approximately 45.9 μs. The RF signal collapse suggests the occurrence of the IMT. Under a large d$H$/d$t$, as produced by the EMFC system in the above-100 T region, when the sample suddenly becomes conductive during a first-order IMT, significant local eddy currents induced by the large d$H$/d$t$ at higher fields cause critical damage to it. As shown in the field dependence of the phase rotation in Fig. 2g, the field-induced IMT occurs at a higher field of ~320 T in addition to the field-induced MIT at ~220 T.

## Discussion

We performed further resistivity and impedance measurements and constructed the $H$-$T$ phase diagram for the $(DMe\text{-}DCNQI)_2Cu$ family, as shown in Fig. 3a (see Supplementary Materials for raw data and a discussion of error bars). Notably, the Pomeranchuk effect is observed within a specific range of magnetic fields. In the low-temperature region of the $x = 0.3$ salt, the slope of the phase boundary exhibits d$H$/d$T < 0$ between 40 T and 120 T, corresponding to the emergence of the reentrant metallic state (Fig. 2e). The slope changes sign at around 40 K, where the entropy relationship between insulating and metallic states is reversed (Fig. 1c). This indicates that the transition is driven by the excess entropy of the insulating state, representing a manifestation of the Pomeranchuk effect.

The underlying entropy behaviour can be explained as follows. Upon the transition to the insulating state, a three-fold periodic ordering of Cu ions with lattice distortion ($...Cu^{2+}Cu^+Cu^+...$) occurs[15,18,22], quenching the π-d hybridisation and forming an energy gap at the Fermi level. The long distance between magnetic $Cu^{2+}$ ions significantly suppresses magnetic interactions, rendering the insulating state paramagnetic. The weak magnetic interactions (Curie-Weiss temperature $\theta$ of −16 K (Fig. S3a in Supplementary Materials)) leads to a high spin entropy arising from disordered $Cu^{2+}$ moments. Consequently, this degenerate spin entropy is released at low temperatures[19], causing the entropy of the insulating state to exceed that of the metallic state. When a magnetic field above 120 T is applied, however, the Pomeranchuk effect is quenched. In the paramagnetic insulating state, entropy is released at higher temperatures under a higher magnetic field, following a two-level Schottky-like behaviour

with an energy gap opened by the Zeeman effect. By contrast, in the metallic state, $S_{met}$ is proportional to the density of states due to Fermi degeneracy, and its variation is negligible compared with the Schottky-like field dependence of $S_{ins}$. As a result, the relationship between $S_{met}$ and $S_{ins}$ reverses at high fields, leading to the disappearance of the Pomeranchuk effect.

As the entropy relationship between metallic and insulating states is reversed above 40 K ($S_{met} > S_{ins}$), the phase boundary turns to d$H$/d$T > 0$. This behaviour is also observed in the $x = 0.5$ and 1 salts, which correspond to the higher-pressure regime of (DMe-DCNQI)$_2$Cu. Here, we focus on the slopes of these phase boundaries, which contrast with typical MITs, where the boundary typically satisfies d$H$/d$T < 0$[23,24]. In conventional electron systems, exchange interactions reduce the magnetic susceptibility of the insulating state compared with the Pauli paramagnetism of the metallic state. Thus, typical insulating states with both lower entropy and lower magnetisation (namely, $S_{met} > S_{ins}$ and $M_{met} > M_{ins}$) are suppressed by an applied magnetic field, leading to d$H$/d$T < 0$ in accordance with the Clausius-Clapeyron relation. In contrast, together with $S_{met} > S_{ins}$, the observed d$H$/d$T > 0$ relationship originates from the π-d hybridisation at the Fermi level[24,25], which gives rise to the positive magnetisation jump at the MIT (namely, $S_{met} > S_{ins}$ and $M_{met} < M_{ins}$)[14,18,22,25–27]. In the metallic state, this π-d hybridisation yields a three-dimensional Fermi liquid state[20]. While the metallic state follows Pauli paramagnetism due to Fermi degeneracy, the paramagnetic insulating state magnetises following the Brillouin function, resulting in the observed $M_{met} < M_{ins}$ relation.

Our results for the $x = 1$ salt indicate that the metallic state becomes stabilised again at ultrahigh fields, as shown in Fig. 3a. This observation suggests that in the ultrahigh-field region, the magnetisation of the metallic state is expected to exceed that of the insulating state. Considering the behaviour of the magnetisation curve, the magnetisation of the paramagnetic insulator state ($M_{ins}$) follows the Brillouin function and approaches saturation in the high-field limit, as illustrated in Fig. 3b. For (DMe-DCNQI)$_2$Cu, the magnetism in the insulating state is solely governed by the $Cu^{2+}$ spins, which account for only one-third of the total Cu population[14,15,26]. Consequently, the saturation value is $\mu_B/3$ per mole (see Fig. S3b in SM). In contrast, in the metallic state, both the $d$-electrons of Cu and the π-electrons of DMe-DCNQI contribute to the Pauli paramagnetism, leading to a magnetisation that exceeds $\mu_B/3$ in the high-field limit. Given the electron bandwidth of approximately 1 eV[28–30], the applied fields in this study remained far below this scale. Therefore, the Fermi liquid state is expected to exhibit linear magnetisation characteristic of Pauli paramagnetism (red curve in Fig. 3b) even at 600 T ($M_{met} \sim \chi_P H$, where $\chi_P$ represents the Pauli paramagnetism). This linear response, which does not saturate at $\mu_B/3$, allows the metallic state to overcome the energy difference $\Delta G$ again (shaded region). Based on these assumptions, $S_{met} = \gamma T$ (Sommerfeld electronic specific heat), $M_{met} = \chi_P H$ (Pauli paramagnetism), $S_{ins} = R[\ln(2\cosh(\mu_B\mu_0 H/k_B T)) + \mu_B\mu_0 H\tanh(\mu_B\mu_0 H/k_B T)/k_B T]$ (Schottky entropy), and $M_{ins} = N_A g\mu_B\tanh(\mu_B\mu_0(H+H_{int})/k_B T)$ (Brillouin function), where $\gamma$, $\mu_B$, $\mu_0$, $k_B$, $N_A$, $g$, and $H_{int}$ represent the Sommerfeld coefficient, Bohr magneton, magnetic constant, Boltzmann constant, and Avogadro constant, $g$-factor, and internal field, respectively, we simulated the phase diagram phenomenologically (see Supplementary Materials for details), as indicated by the dashed curve in Fig. 3a. The simulation aligns well with the experimental data. Notably, this simulation assumes a simplified two-state model (liquid and solid), neglecting potential intermediate phases. If such phases, like liquid crystals of Pomeranchuk electrons, emerge at high fields, the experimental results should deviate from the simulation. The slight discrepancy of the data point of 109 K (310 T) might be related to the presence of such phases. However, due to technical constraints in the EMFC experiments, future theoretical studies are required for deeper discussion.

Finally, we consider the analogy of our results with $^3$He. Just as the field-induced solidification in (DMe-DCNQI)$_2$Cu, the same phenomenon is also expected to occur in $^3$He, inherently implying the quenching of the Pomeranchuk effect at sufficiently high magnetic fields. The field-induced IMT in (DMe-DCNQI)$_2$Cu can also be interpreted as a consequence of molecular orbital dissociation under a strong magnetic field. When a magnetic field destabilises the antiparallel spin configuration, the system undergoes IMT. This perspective suggests a similar mechanism in $^3$He: if the magnetic field energy surpasses the binding energy responsible for the formation of $^3$He solid—namely, van der Waals interactions—a high-field liquid state of $^3$He might emerge. In the solid state, $^3$He atoms exhibit substantial zero-point vibrations even at low temperatures, with amplitudes reaching approximately 40% of the lattice spacing[31]. Recent thermal Hall effect measurements[32,33] suggest that lattice vibrations couple directly to a magnetic field, potentially suppressing van der Waals interactions in ultrahigh fields. Such a non-trivial lattice collapse could lead to a novel quantum liquid state distinct from the conventional liquid state of $^3$He, presenting an intriguing avenue for future research.

In summary, we revisited the electronic states of [(DMe-DCNQI)$_{1-x}$($d_8$-DMe-DCNQI)$_x$]$_2$Cu, which exhibits the Pomeranchuk effect closely resembling that of $^3$He. Using [(DMe-DCNQI)$_{1-x}$($d_8$-DMe-DCNQI)$_x$]$_2$Cu, we successfully traced the behaviour of the Pomeranchuk effect in the high-field limit, observing its quenching in high fields and reentrant melting under extreme magnetic fields. From a phenomenological perspective, we demonstrated that this extraordinary field dependence can be understood through the relationship between the magnetisation and entropy of the Pomeranchuk electrons. By analogy to $^3$He, $^3$He may also exhibit a unique sequence of magnetic-field-induced solidification and remelting in the high-field limit.

## Methods
### Sample preparation
Single crystals of [(DMe-DCNQI)$_{1-x}$($d_8$-DMe-DCNQI)$_x$]$_2$Cu were synthesised by the slow chemical reduction of mixture of DMe-DCNQI and d8-DMe-DCNQI using CuI in an acetonitrile solution. The mixing ratio $x$ is determined by the ratio of the mixture DMe-DCNQI and d8-DMe-DCNQI. The typical size of the obtained crystals is about 50 μm × 50 μm × 1000 μm.

### Electrical transport measurements in pulsed magnetic fields
Electrical transport measurements in pulsed magnetic fields were performed by typical ac methods below 100 T and radio-frequency impedance method above 100 T[34].

### Generation of pulsed magnetic fields
Pulsed magnetic fields were generated using in-house pulsed magnets at the Institute for Solid State Physics, University of Tokyo. For magnetic fields below 100 T, two types of non-destructive pulsed magnets were employed. The first generated a maximum field of 60 T with a pulse duration of 36 ms, while the second achieved 88 T with a pulse duration of 3 ms. For the generation of magnetic fields exceeding 100 T, destructive pulsed magnets, single-turn coils (STCs) and electromagnetic flux compression (EMFC) systems[21], were utilised. The STCs generated pulsed magnetic fields ranging from 135 to 260 T within 4−6 μs, while the EMFC system achieved fields as high as over 600 T. Detailed magnetic field profiles used in this study are provided in the Supplementary Materials.

## Data availability
Data supporting the findings of this study and its supplementary information files are available from the corresponding authors upon request.

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

## Acknowledgements

We thank H. Sawabe, H. Takahashi, and X. Zhou for technical support. A part of this work was supported by JSPS KAKENHI (Grant No. 24K17005 (S.I.), 24H01605 (S.I.), 23H04859 (Y.H.M.), and 23H04860 (Y.H.M.)), and JP22H00106 (H.M.), Core-to- Core Program "Emergent Quantum Electronics in Molecular Layer" (JPJSCCA20240001 to S.I, S.Y. and H.M.).

## Author contributions

N.M., H.I., K.M., K.K. and S.I. performed the measurements using non-destructive pulsed magnets. N.M., T.N., Y.I., H.H., Y.H.M. and S.I. performed the measurements using destructive pulsed magnets. S.Y. and H.M. provided single crystals used in this study. S.I. conceived and supervised the project. N.M. and S.I. wrote the paper.

## Competing interests

The authors declare no competing interests.
