## [Peer Review File · Nature Communications]

Fate of Pomeranchuk effect in ultrahigh magnetic fields

Corresponding Author: Professor Shusaku Imajo

Version 0:

Reviewer comments:

Reviewer #1

(Remarks to the Author)

The work by N. Matsuyama et al. presents ultrahigh magnetic field characterization of an interesting organic conductor, where field induced metal-insulator transitions and the unusual relationship between critical fields and temperatures have been established. The results support the explanation of entropy driven Pomeranchuk effect, analogous to that happens in ^3He . The experiment is intriguing and the physics unveiled is significant. I recommend its publication after some revisions.

1. The material studied should be described clearly at the beginning of the main text, at least in the last paragraph of introduction. Besides the basic information about crystal structures and electronic bands (mentioned a little in the second paragraph in Discussion), the authors should introduce in brief several kinds of explanations of the MIT and reentrant metallic state. In particular, one of them (ref. 15) is highly relevant in regards of entropy measurements. This work is an independent examination of the explanation of Pomeranchuk effect, which should be presented in a clearer background of recent progresses.
2. The minimum transition B field for $x=0.3$ is found at 40 K. Is this temperature related to the extreme point of the melting curve in Fig. 1b? The hysteresis in Fig. 2b-d should be discussed in details. Can more information be inferred?
3. The measurement of $x=0.45$ that have direct observation of reentrant metallic states in the low temperature regime has been in lack. Any low temperature limitation for the high field measurement? It will be a strong connection between the presented two ends, i.e., $x=1$ and 0.3. In addition, the data points in Fig. 3 seem to be more than those in Fig. 2. The raw data should be added in SI.
4. The manuscript repeatedly refers to the analogy with ^3He across multiple sections, leading to redundancy and dilution of the key message. Conversely, important aspects such as the modeling behind Fig. 3b are presented rather briefly, without adequate explanation of the assumptions or derivations. A more compact and focused structure—concentrating on the core physics rather than reiterating the heuristic analogy—would benefit the clarity and impact of the work.
5. While the manuscript is generally well-written and grammatically correct, several aspects of the scientific writing could benefit from clarification and greater precision. For instance, the phrase “an anomalous entropy relation” appears repeatedly but is not clearly defined. It would help to explicitly state the direction of the entropy difference between phases to avoid ambiguity. Additionally, in several instances, key modeling assumptions (e.g., specific forms of heat capacity or magnetization) are introduced only implicitly. Expanding these descriptions would enhance the transparency and reproducibility of the theoretical analysis.

Overall, the work deserves to be published in Nature Communications after properly addressing the above concerns.

Reviewer #2

(Remarks to the Author)

Reviewer #3

(Remarks to the Author)
Attached

Reviewer #4

(Remarks to the Author)
I co-reviewed this manuscript with one of the reviewers who provided the listed reports. This is part of the Nature Communications initiative to facilitate training in peer review and to provide appropriate recognition for Early Career Researchers who co-review manuscripts.

Version 1:

Reviewer comments:

Reviewer #1

(Remarks to the Author)
The authors have fully addressed my concerns. Now I recommend its publication in the current form.

Reviewer #2

(Remarks to the Author)
I co-reviewed this manuscript with one of the reviewers who provided the listed reports. This is part of the Nature Communications initiative to facilitate training in peer review and to provide appropriate recognition for Early Career Researchers who co-review manuscripts.

Reviewer #3

(Remarks to the Author)
In the response letter and revised version of the manuscript, the authors have thoroughly responded to my previous comments. They have improved the text and figures to avoid confusion. The additionally measurements on the $x = 0.5$ salt at high temperatures provide further insight into the field response of the system, supporting the connection between the $x = 0.3$ and $x = 1$ salts. Now, I recommend this manuscript for publication.

Reviewer #4

(Remarks to the Author)
I co-reviewed this manuscript with one of the reviewers who provided the listed reports. This is part of the Nature Communications initiative to facilitate training in peer review and to provide appropriate recognition for Early Career Researchers who co-review manuscripts.

Fate of Pomeranchuk effect in ultrahigh magnetic fields

Naofumi Matsuyama*, So Yokomori, Toshihiro Nomura, Yuto Ishii, Hiroaki Hayashi, Hajime Ishikawa, Kazuki Matsui, Hatsumi Mori, Koichi Kindo, Yasuhiro H. Matsuda, and Shusaku Imajo*

We thank the Reviewers for their insightful comments on our paper, which have substantially elevated the quality of our manuscript. In response to their valuable feedback, we have implemented the following revisions, which are intended to address the Reviewers' concerns. We thank Reviewers #2 and #4 for their time and effort in co-reviewing our manuscript.

Response to Reviewer #1

The work by N. Matsuyama et al. presents ultrahigh magnetic field characterization of an interesting organic conductor, where field induced metal-insulator transitions and the unusual relationship between critical fields and temperatures have been established. The results support the explanation of entropy driven Pomeranchuk effect, analogous to that happens in 3He. The experiment is intriguing and the physics unveiled is significant. I recommend its publication after some revisions.

[reply] We thank the first Reviewer for rating our work highly and raising valuable suggestions that are helpful to improve our manuscript. We reply point by point to all the comments below and indicate the revised parts of the manuscript.

[Comment 1] The material studied should be described clearly at the beginning of the main text, at least in the last paragraph of introduction. Besides the basic information about crystal structures and electronic bands (mentioned a little in the second paragraph in Discussion), the authors should introduce in brief several kinds of explanations of the MIT and reentrant metallic state. In particular, one of them (ref. 15) is highly relevant in regards of entropy measurements. This work is an independent examination of the explanation of Pomeranchuk effect, which should be presented in a clearer background of recent progresses.

[reply] We thank the Reviewer for their helpful advice. As pointed out, it is important to clarify the present material at the beginning of the manuscript. We have revised the introduction to include a more detailed explanation of the fundamental properties of the materials, as well as the MIT and the re-entrant metallic behaviour. In addition, as the Reviewer noted, Ref. 15 is an essential prior study for the discussion of entropy. We have now carefully incorporated it into the revised version as part of the background of the advances presented in this work.

[Change]

(line 103) “One-dimensional DMe-DCNQI columns coordinated by Cu ions form a tetragonal lattice of space group $I4_1/a$, showing metallic behaviour down to the lowest temperatures arising from a hybridised electron band composed of DCNQI π -electrons and Cu d-electrons. External pressure or chemical substitution can shift the ground state into a Cu^{2+} -originated paramagnetic insulator accompanied by a three-fold superlattice formation, with an intermediate range where a reentrant metal-insulator-metal transition is observed (Fig. 1d)”

(line 110) “While an earlier study has investigated the entropic relation between metallic and insulating states, which explains the emergence of the reentrant metallic phase, its link to the Pomeranchuk effect remains unexplored.”

[Comment 2] The minimum transition B field for $x=0.3$ is found at 40 K. Is this temperature related to the extreme point of the melting curve in Fig. 1b? The hysteresis in Fig. 2b-d should be discussed in details. Can more information be inferred?

[reply] We thank the Reviewer for the insightful comment. As the Reviewer pointed out, the minimum in the transition field at 40 K is related to the minimum in the melting curve on the P – T phase diagram shown in Fig. 1b. At 40 K, the entropies of the liquid and solid states become equal, resulting in a zero slope of the melting curve according to the Clausius–Clapeyron relation, since the transition entropy ΔS is zero at this temperature (see the top panel of Fig. 1b).

Consequently, the minimum transition field also appears at 40 K in the H – T phase diagram. This correspondence provides strong support for the thermodynamic interpretation of our system.

Regarding the observed hysteresis in Figs. 2b–d, it is a characteristic feature of nucleation and growth in first-order phase transitions. Although the transition is very sharp, indicating rapid domain growth, nucleation requires overcoming an energy barrier, which leads to the observed hysteresis. In the revised manuscript, we have added a brief explanation. We note that a detailed discussion of hysteresis is challenging because the field-sweep rate at the phase transition differs among the measurement datasets.

[Change]

(line 146) “The phase boundary, shown in Fig. 2d, exhibits a minimum in the transition field H_{MIT} around 40 K, which corresponds to the temperature at which the entropies of the insulating and metallic states become equal (see Fig. 1c)”

(line 140) “The field-induced MITs are first-order phase transitions accompanied by hysteresis. The wide hysteresis indicates that nucleation associated with the phase transition requires overcoming an energy barrier, whereas the sharpness of the transition implies that domain growth develops rapidly.”

[Comment 3] The measurement of $x=0.45$ that have direct observation of reentrant metallic states in the low temperature regime has been in lack. Any low temperature limitation for the high field measurement? It will be a strong connection between the presented two ends, i.e., $x=1$ and 0.3 . In addition, the data points in Fig. 3 seem to be more than those in Fig. 2. The raw data should be added in SI.

[reply] Our samples for $x = 0.45$ were unfortunately thin and fragile, which prevented us from performing pulsed-field measurements at low temperatures. As the temperature was decreased, the sample underwent two MITs, until it cracked. It is indeed unfortunate that the field effect on re-entrant metallic states is lacking. However, the electronic state of re-entrant metallic state has been shown to possess no difference compared with the metallic state seen in high-temperature or low-pressure regions (*Solid State Commun.* **93**, 203 (1995).). Thus, we do not think the lack of the low-temperature data for $x = 0.45$ weakens our discussion.

Whereas we could not investigate the low-temperature region with our $x = 0.45$ sample, we succeeded in conducting additional measurements on an $x = 0.5$ salt at high temperatures. We believe these data provide further information on the field response of the system, supporting the connection between the $x = 0.3$ and $x = 1$ salts. In addition, we have added Fig. 2e, which shows the 40 T data for $x = 0.3$, clearly indicating a similarity to the $x = 0.45$ data at 0 T. Therefore, the $x = 0.45$ data have been moved into the inset of Fig. 2e, and the $x = 0.5$ data has been added to Figs. 2a and 3a.

As for the data points shown in the phase diagram (Fig. 3a), we have summarised the raw data in another section of our Supplementary Materials.

[Change]

(Fig. 2a, 2e, and 3a)

(Fig. S4 in SM) We added a new section 5 in our Supplementary material.

[Comment 4] The manuscript repeatedly refers to the analogy with ³He across multiple sections, leading to redundancy and dilution of the key message. Conversely, important aspects such as the modeling behind Fig. 3b are presented rather briefly, without adequate explanation of the assumptions or derivations. A more compact and focused structure—concentrating on the core physics rather than reiterating the heuristic analogy—would benefit the clarity and impact of the work.

[reply] We appreciate this comment as well. We have revised our manuscript to include a more detailed explanation of our simulation, which was previously mentioned in our Supplementary Material. Additionally, we have separated the discussion on the experimental results on (DMe-DCNQI)₂Cu systems and shortened the part discussing the analogy to ³He.

[Change]

(line 227) “Based on these assumptions, $S_{\text{met}} = \gamma T$ (Sommerfeld electronic specific heat), $M_{\text{met}} = \chi_{\text{p}} H$ (Pauli paramagnetism), $S_{\text{ins}} = R[\ln(2\cosh(\mu_{\text{B}}\mu_0 H/k_{\text{B}}T)) + \mu_{\text{B}}\mu_0 H \tanh(\mu_{\text{B}}\mu_0 H/k_{\text{B}}T)/k_{\text{B}}T]$ (Schottky entropy), and $M_{\text{ins}} = N_{\text{A}}g\mu_{\text{B}}\tanh(\mu_{\text{B}}\mu_0(H+H_{\text{int}})/k_{\text{B}}T)$ (Brillouin function), where γ , μ_{B} , μ_0 , k_{B} , N_{A} , g , and H_{int} represent the Sommerfeld coefficient, Bohr magneton, magnetic constant, Boltzmann constant, and Avogadro constant, g -factor, and internal field, respectively, we simulated the phase diagram phenomenologically (see Supplementary Materials for details), as indicated by the dashed curve in Fig. 3a.”

(The paragraph starting at line 239 has been shortened.)

[Comment 5] While the manuscript is generally well-written and grammatically correct, several aspects of the scientific writing could benefit from clarification and greater precision. For instance, the phrase “an anomalous entropy relation” appears repeatedly but is not clearly defined. It would help to explicitly state the direction of the entropy difference between phases to avoid ambiguity. Additionally, in several instances, key modeling assumptions (e.g., specific forms of heat capacity or magnetization) are introduced only implicitly. Expanding these descriptions would enhance the transparency and reproducibility of the theoretical analysis.

[reply] We appreciate the Reviewer’s comments on the ambiguity in our original manuscript. We revised our Discussion section, especially taking special care to eliminate ambiguous terms.

[Change]

(line 199) “($S_{\text{met}} > S_{\text{ins}}$)”, (line 205) (namely, $S_{\text{met}} > S_{\text{ins}}$ and $M_{\text{met}} > M_{\text{ins}}$), (line 207) $S_{\text{met}} > S_{\text{ins}}$, (line 209) (namely, $S_{\text{met}} > S_{\text{ins}}$ and $M_{\text{met}} < M_{\text{ins}}$)

(line 225) “ $M_{\text{met}} \sim \chi_{\text{p}} H$, where χ_{p} represents the Pauli paramagnetism”

(line 227) “Based on these assumptions, $S_{\text{met}} = \gamma T$ (Sommerfeld electronic specific heat), $M_{\text{met}} = \chi_{\text{p}} H$ (Pauli paramagnetism), $S_{\text{ins}} = R[\ln(2\cosh(\mu_{\text{B}}\mu_0 H/k_{\text{B}}T)) + \mu_{\text{B}}\mu_0 H \tanh(\mu_{\text{B}}\mu_0 H/k_{\text{B}}T)/k_{\text{B}}T]$ (Schottky entropy), and $M_{\text{ins}} = N_{\text{A}}g\mu_{\text{B}}\tanh(\mu_{\text{B}}\mu_0(H+H_{\text{int}})/k_{\text{B}}T)$ (Brillouin function), where γ , μ_{B} , μ_0 , k_{B} , N_{A} , g , and H_{int} represent the Sommerfeld coefficient, Bohr magneton, magnetic constant, Boltzmann constant, and Avogadro constant, g -factor, and internal field, respectively, we simulated the phase diagram phenomenologically (see SM for details), as indicated by the dashed curve in Fig. 3a.”

[Comment 6] Overall, the work deserves to be published in Nature Communications after properly addressing the above concerns.

[reply] We appreciate all the insightful comments from the Reviewer and believe our revised manuscript is now ready for publication in Nature Communications.

Response to Reviewer #3

In this manuscript, the authors revisit the electronic properties of the coordination polymer [(DMe-DCNQI)_{1-x}(d8-DMe-DCNQI)_x]₂Cu for $x = 0.3, 0.45,$ and 1 through the analogy with Pomeranchuk effect in He³. According to the data shown in the manuscript, the Pomeranchuk effect indeed appears for the $x = 0.45$ salt at zero field, and for the $x = 0.3$ salt within the range $\sim 30 \text{ T} < \mu_0 H < \sim 120 \text{ T}$. However, for the $x = 1$ salt, no evidence supports the Pomeranchuk effect based on the standard definition (liquid system solidifies upon heating) and they interpret the ultrahigh magnetic field data as evidence of a re-entrant liquid state. This behavior is phenomenologically demonstrated using the relationship between magnetization and entropy of the electrons. However, based on the data and discussions presented in the current form, I can not suggest publication.

[reply] We sincerely appreciate the Reviewer's thorough review and constructive comments. We agree that the concerns raised by the Reviewer are essential for the proper interpretation of our results. Following the suggestions, we have revised the manuscript to clarify our interpretation. We hope that these revisions will facilitate a better understanding of our findings.

As Reviewer #3 pointed out, the $x = 1$ salt itself does not exhibit the Pomeranchuk effect. However, given its continuous connection in the phase diagram with the $x = 0.3$ and 0.45 salts, which do exhibit the Pomeranchuk effect, the results for the $x = 1$ salt are also important for a comprehensive understanding of an "electron system exhibiting the Pomeranchuk effect." This continuous connection is supported by a past dHvA experiment, in which the re-entrant metallic state has been shown to possess no difference compared with the metallic state observed in the high-temperature or low-pressure region (*Solid State Commun.* **93**, 203 (1995)).

To solidify the discussion of this connection, we additionally conducted measurements on the $x = 0.5$ salt at high temperatures, as it was challenging to investigate our $x = 0.45$ salt. Indeed, the field response shows a continuous change with the variation in the fraction x . We believe these data provide further information on the field response of the system, supporting the connection between the $x = 0.3$ and $x = 1$ salts.

Regarding the evidence for a re-entrant liquid state in the $x = 1$ salt, we have explained in detail why our interpretation is correct and why the evidence is sufficient to conclude the re-entrant behaviour in our reply to [Comment 1].

[Change]

(Fig. 3a) We have added the $x = 0.5$ data.

[Comment 1] The authors report their one remarkable finding that the $x = 1$ salt displays a re-entrant liquid state in ultrahigh magnetic fields. This conclusion is hardly supported due to the excessively noisy high-field data, making it difficult to distinguish the IMT signal from the noise. I also have concerns regarding magnetic field induced heating effect. The manuscript is presented assuming the sample temperature is fixed during the measurements. For sure it is very challenging to conduct experiments under this high magnetic field. However, it is highly possible that the sample is heated up during the rapid field ramping up process. Then "re-entrant metallic state" is just the high-temperature phase and not very interesting. Consequently, the title of the manuscript, "Fate of Pomeranchuk effect in ultrahigh magnetic fields," is misleading.

[reply] We appreciate the comment pointing out the unclear description of the noisy data and the heating effect in the previous manuscript.

We here reply to the question regarding the observed noise in the EMFC measurements. In single-turn coil measurements, significant noise is often observed due to the electromagnetic noise produced by the high-voltage switching of capacitors. However, we believe that the only possible origin of the noise is sample destruction in the EMFC measurements, based on two main reasons.

First, the data before 45.3 μs exhibit no significant noise, and the noise starts at 45.3 μs , before the coil explosion at 45.9 μs , which indicates that the noise is unrelated to the collapse of the experimental setup due to the explosion. Second, EMFC measurements are typically tolerant of extrinsic noise before the coil explosion (e.g. see *Nature Commun.* **14**, 1744 (2023)., *Phys. Rev. B* **105**, L251105 (2022)). Although electrical measurements using destructive single-turn coils are inherently challenging and the data obtained are often noisy, in EMFC measurements the sample space remains electrically stable and quiet, with minimal mechanical vibration, until the explosion of the coil, because dH/dt is extremely slow at the initial stage of field generation. Once the sample is destroyed, however, the circuit suddenly becomes open, causing the open terminal to act as an antenna collecting electrical noise. Consequently, the phase of the signal becomes unstable, and noise is expected to occur.

As for the mechanism of sample destruction, we speculate that eddy-current heating induced by the time-varying magnetic field would destroy our sample together with the sharp first-order IM transition. The non-uniform growth of metallic domains during the transition causes an uneven distribution of eddy currents and resulting local heating, eventually destroying the sample. For the MI transition in the low-field region, we were able to measure the resistance without sample breakage. Since the insulating state has much higher resistivity, insulating domain growth would not lead to an increase in eddy currents.

Next, we would like to respond to the second concern regarding the heating effect. As discussed above, we also considered heating as a possible source of sample destruction. The estimated temperature increase during the period when the sample is metallic is around 10 K (see Section 4 in Supplementary Material for details). However, since the insulating state, with eight orders of magnitude higher resistivity, would be free from heating by eddy currents, we assume that the isothermal condition holds in the insulating phase, namely the field range of 220-340 T. Therefore, we can conclude that the re-entrant phase originates not from heating, but from the magnetic-field-induced transition.

As for the concern on the sample temperature, the details are described in our Supplementary Materials (see the section .4).

[Change]

(line 170) “Under a large dH/dt , as produced by the EMFC system in the above-100 T region, when it suddenly becomes conductive during a first-order IMT, significant local eddy currents induced by the large dH/dt at higher fields cause critical damage to the sample.”

(Section 4 in Supplementary Materials) “It should be noted that the sample breakage originates from heating induced by metallisation during the insulating–metal transition. If the phase transition proceeded uniformly, it would not result in breakage, as the discussion above indicates that only heating on the order of 10 K would occur. However, the phase transition in this system is first-order and accompanied by a large hysteresis. When a metallic domain begins to form through nucleation, the current density becomes concentrated within that domain, generating a substantial local heating effect. Damage to this region then causes the current density to increase further in the next domain undergoing metallisation, ultimately leading to sample breakage.”

[Comment 2] According to the authors, the metal-insulator transition for $x=1$ sample is accompanied by a structural phase transition. While here, they want to claim a purely electronic Pomeranchuk effect. This needs to be more clearly addressed.

[reply] Not only for the $x = 1$ sample, the transition between metallic and insulating states in (DMe-DCNQI)₂Cu systems is accompanied by a structural transition. The three-fold superlattice formation is a common characteristic of the insulating state of DCNQI–Cu salts (*Bull. Chem. Soc. Jpn.*, **73**, 515 (2000)). We constructed our discussion assuming the metallic and insulating states, which can be switched with each other by changing the external magnetic field. Depending on the strength of the external magnetic field, the favoured state is either metallic or insulating, which is reproduced based on our thermodynamic simulation shown in the manuscript. Magnetic fields can directly affect the spin degree of freedom, which acts as a driving force for the transition between metallic and insulating states. Since the structural phase transition is a resultant accompaniment of the electronic transition and is not the essence of this phase transition, the main logic would not be altered by including a structural phase transition.

While the discussion itself is preserved in its original form, to clarify the fundamental properties of the investigated system, we have revised our introduction. We hope this resolves the Reviewer's concern.

[Change]

(line 103) “One-dimensional DMe-DCNQI columns coordinated by Cu ions form a tetragonal lattice of space group $I4_1/a$, showing metallic behaviour down to the lowest temperatures arising from a hybridised electron band composed of DCNQI π -electrons and Cu d-electrons. External pressure or chemical substitution can shift the ground state into a Cu²⁺-originated paramagnetic insulator accompanied by a three-fold superlattice formation, with an intermediate range where a re-entrant metal-insulator-metal transition is observed (Fig. 1d)”

[Comment 3] What is the magnetic response for the $x = 0.45$ salt, which shows the re-entrant metallic state at low temperatures?

[reply] Our samples for $x = 0.45$ were unfortunately thin and fragile, which prevented us from performing pulsed-field measurements at low temperatures. As the temperature was decreased, the sample underwent two MITs, until it cracked. It is indeed unfortunate that the field effect on re-entrant metallic states is lacking. However, the electronic state of re-entrant metallic state has been shown to possess no difference compared with the metallic state seen in high-temperature or low-pressure regions (*Solid State Commun.* **93**, 203 (1995)). Thus, we do not think the lack of the low-temperature data for $x = 0.45$ weakens our discussion.

Whereas we could not investigate the low-temperature region with our $x = 0.45$ sample, we succeeded in conducting additional measurements on an $x = 0.5$ salt at high temperatures. We believe these data provide further information on the field response of the system, supporting the connection between the $x = 0.3$ and $x = 1$ salts. In addition, we have added Fig. 2e, which shows the 40 T data for $x = 0.3$, clearly indicating a similarity to the $x = 0.45$ data at 0 T. Therefore, the $x = 0.45$ data have been moved into the inset of Fig. 2e, and the $x = 0.5$ data has been added to Figs. 2a and 3a.

[Change]

(line 130) “For the fully deuterated salt ($x = 1$) and $x = 0.5$ salts, sharp resistance jumps at $T \sim 80$ K and 60 K, respectively.”

(line 149) “As illustrated in Fig. 2e, the $x = 0.3$ salt is metallic throughout the temperature range at 0 T. However, under a high magnetic field of 40 T, it exhibits reentrant MIM transitions, similar to the behaviour observed for the $x = 0.45$ salt at 0 T (inset of Fig. 2e).”

(Fig. 2a, 2e, and 3a)

[Comment 4] Other minor suggestions and points needed to be addressed:

1. No page number is labeled.
2. Page 3, “Only a few examples of electrons exhibiting the Pomeranchuk like effect in solids have been reported”. In recent years, the Pomeranchuk-like effect has been observed in more 2D systems, including TMDCs moiré systems (Nature 597, 350-354 (2021), Phys.Rev. X 12, 041015 (2022),) and rhombohedral multilayer graphene system (Nature 640, 355–360 (2025)).
3. Page 4, “Chemical pressure, the counterpart of physical pressure in 3He, was introduced by mixing deuterated DMe-DCNQI (d8-DMe-DCNQI)”, How does the chemical pressure P depend on the d8-DMe DCNQI fraction x ?
4. In Page 6, “In the lower-temperature region of the $x = 0.3$ salt, the slope of the phase boundary exhibits $dH/dT < 0$, coinciding with the emergence of the reentrant metallic state.” This is a bit confusion, coinciding with the reentrant metallic state in $x=0.45$ salt?
5. In Figure 2a, why is the ρ - T curve not smooth for $x = 0.45$?
6. In Figure 2b, 2c, and 2d, what do the light-colored curves represent? It does not seem to have been mentioned anywhere in the manuscript. It’s likely the down-ramped curve. Would the hysteresis cause by temperature difference? Why aren't the lighter-colored curves shown for the two middle panels of Figure 2c?
7. Page 4, “These results indicate that the region of the insulating phase in the pressure temperature phase diagram expands toward the lower-pressure side under magnetic fields”, authors could draw a schematic diagram for this statement.
8. Page 5, “Figure 2d illustrates the relative phase change $\Delta\phi$ of RF waves as a function of the magnetic field at 100 K and 110 K”, the temperatures labeled in Figure 2d are 98 K and 107 K, respectively.
9. In Figure 3a, the authors should state how the data uncertainties were determined.
10. The authors interpret the noise in Fig. 2f and 2g as the damage to the sample caused by eddy currents. Is there further evidence of sample damage? Is it feasible to reduce dH/dt ? Do the authors have the data for decreasing magnetic field conditions in the EMFC system?

[reply]

1. We put the page numbers in our manuscript.
2. We thank the Reviewer for bringing up the latest research on the Pomeranchuk effect observed in 2D materials. We have added the references to our manuscript and revised the text accordingly. In contrast to these 2D systems, we believe that the coordination polymer, in which the Pomeranchuk effect arises from π - d hybridization, constitutes a new class of materials worthy of investigation.
3. We have revised the introduction to provide more fundamental information about the investigated materials. As mentioned there, deuterium substitution corresponds to positive chemical pressure (*Bull. Chem. Soc. Jpn.* **73**, 515 (2000)).
The effective pressure arising from selective deuteration of the methyl group in (DMe-DCNQI)₂Cu salts was first reported to increase linearly with the number of deuterium atoms in each methyl group. Later, it was revealed that alloy systems of (DMe-DCNQI)₂Cu and the fully deuterated (d8-DMe-DCNQI)₂Cu can also reproduce the low-pressure region of the P - T phase diagram of the (DMe-DCNQI)₂Cu system, including the region where insulating and re-entrant metallic ground states are obtained (see *Solid State Commun.* **85**, 831-835 (1993)). We have also revised Fig. 1d to better highlight the chemical pressure arising from the alloy effect.

4. We intended to highlight that the Pomeranchuk effect is observed in the H - T phase diagram in the low-temperature regime, where $dH/dT > 0$. However, as the Reviewer pointed out, the discussion in the original form was somewhat confusing. We have revised our manuscript to avoid this confusion and make the discussion more straightforward.

5. Indeed, the response of $x = 0.45$ does not appear smooth, showing a gradual increase before the discontinuous jump. A similar gradual increase was also observed in $x = 1$ salts, although it is not clearly visible in the figure. This characteristic can also be seen in past resistivity results (e.g., Fig. 2 in *Solid State Commun.* **85**, 831 (1993)). As the precursor-like behaviour depends on the sample, sample quality, such as inhomogeneity induced by alloying, may play a role; however, the underlying cause of this gradual increase has yet to be elucidated.

6. As Reviewer #3 pointed out, the lightly coloured curves represent the behaviour for the field-descending part. For some curves in Fig. 2c, the hysteresis was too large, and the insulating state survived down to zero field. We have added this explanation to the figure caption.

7. We agree with your suggestion. We also expect that we could illustrate the phase boundary shift, as displayed in Fig. R1. However, in this study we investigated only the samples with $x = 0.3, 0.5,$ and 1 , and thus it is difficult to draw it accurately. Moreover, in the high-field limit, the field-induced re-entrant behaviour should result in a reversed change. Therefore, we have omitted the text to avoid potential misinterpretation.

Fig. R1 Schematic of the P - T Phase boundary at 0 T (blue) and high magnetic fields (red).

8. We thank the Reviewer for pointing out the inconsistency in our notification. We have corrected and made them consistent in the temperature values of these two measurements.

9. The sample temperature has some uncertainty due to eddy-current heating. As discussed in our Supplementary Materials, this heating effect is especially severe in experiments with destructive magnets. Based on the estimated temperature rise during field generation, we included a bar to indicate this uncertainty. To clarify, we added a sentence guiding the readers to the Supplementary Materials for details on the sample heating estimation.

As for the uncertainty in the magnetic field value in EMFC experiments, it arises from the possible deviation of the sample from the coil centre. According to past experimental results, a

deviation of only 3 mm can result in a maximum field decrement of ~ 100 T (*Rev. Sci. Instrum.* **85**, 036102 (2014)).

10. Since the whole experimental setup around the coil is destroyed and lost after the EMFC measurements, it is quite challenging to further verify the sample damage. However, in single-turn coil measurements, where the setup can be preserved even after field generation, we indeed had several examples in which the sample was damaged and broken after experiencing rapid dH/dt . These experimental observations empirically support our discussion on sample damage related to the noise in Figs. 2f and 2g.

Regarding the idea of slowing down field generation, this is unfortunately impossible. As shown in Supplementary Material 1, dH/dt shows a steep increase above the 100 T region in EMFC experiments. Although the pulsed field generated by single-turn coils shows a slowing down of dH/dt near the maximum field, the maximum field that single-turn coils can reach is not sufficient to observe the reentrant MIM transition. Finally, as we mentioned in our Supplementary Materials, field-descending data cannot be obtained in EMFC measurements due to the destruction of the setup at the maximum field.

[Change]

2. (Ref 10) Li, T., Jiang, S., Li, L. et al. Continuous Mott transition in semiconductor moiré superlattices. *Nature* **597**, 350-354 (2021).

(Ref. 11) Zhang, M., Zhao, X., Watanabe, K. et al., Tuning Quantum Phase Transitions at Half Filling in 3L-MoTe₂/WSe₂ Moiré Superlattices *Phys.Rev. X* **12**, 041015 (2022).

(Ref. 12) Holleis, L., Xie, T., Xu, S., et al. Fluctuating magnetism and Pomeranchuk effect in multilayer graphene. *Nature* **640**, 355–360 (2025).

3. (line 120) “The deuteration of DMe-DCNQI acts as a positive chemical pressure, as shown in Fig. 1d.”, (Fig. 1d) The right vertical axis.

4. (line 180) “In the lower-temperature region of the $x = 0.3$ salt, the slope of the phase boundary exhibits $dH/dT < 0$ between 40 T and 120 T, corresponding to the emergence of the reentrant metallic state (Fig. 2e).”

6. (line 383) “Dark-coloured curves correspond to data obtained during field-ascending sweeps, while light-coloured curves correspond to data obtained during field-descending sweeps. In some cases, due to large hysteresis, the insulating state persists down to zero field.”

8. (line 157) “Figure 2f illustrates the relative phase change $\Delta\phi$ of RF waves as a function of the magnetic field at 98 K and 107 K, measured in pulsed fields generated by the STC system.”

9. (line 396) “Bars attached to data points represent uncertainties in temperature and magnetic field. Temperature uncertainty arises from heating during a field generation (see Supplementary Material), whereas magnetic field uncertainty reflects the electromagnetic noise in STC experiments (see Supplementary Material for raw data) and possible deviation from the field centre due to sample length in EMFC experiments.”

(Ref. 34) Nakamura, D., Sawabe, H., Takeyama, S. Experimental evidence of three-dimensional dynamics of an electromagnetically imploded liner. *Rev. Sci. Instrum.* **85**, 036102 (2014).

We express our sincere gratitude to the Reviewers for their valuable comments, which have significantly enhanced the quality of our manuscript. We look forward to hearing from all the Reviewers regarding our resubmission and remain available to address any additional queries or remarks they may have.

In this manuscript, the authors revisit the electronic properties of the coordination polymer $[(\text{DMe-DCNQi})_{1-x}(\text{d8-DMe-DCNQi})_x]_2\text{Cu}$ for $x = 0.3, 0.45,$ and 1 through the analogy with Pomeranchuk effect in He^3 . According to the data shown in the manuscript, the Pomeranchuk effect indeed appears for the $x = 0.45$ salt at zero field, and for the $x = 0.3$ salt within the range $\sim 30 \text{ T} < \mu_0 H < \sim 120 \text{ T}$. However, for the $x = 1$ salt, no evidence supports the Pomeranchuk effect based on the standard definition (liquid system solidifies upon heating) and they interpret the ultrahigh magnetic field data as evidence of a re-entrant liquid state. This behavior is phenomenologically demonstrated using the relationship between magnetization and entropy of the electrons. However, based on the data and discussions presented in the current form, I can not suggest publication. The main concerns are as follows:

1. The authors report their one remarkable finding that the $x = 1$ salt displays a re-entrant liquid state in ultrahigh magnetic fields. This conclusion is hardly supported due to the excessively noisy high-field data, making it difficult to distinguish the IMT signal from the noise. I also have concerns regarding magnetic field induced heating effect. The manuscript is presented assuming the sample temperature is fixed during the measurements. For sure it is very challenging to conduct experiments under this high magnetic field. However, it is highly possible that the sample is heated up during the rapid field ramping up process. Then “re-entrant metallic state” is just the high-temperature phase and not very interesting. Consequently, the title of the manuscript, “Fate of Pomeranchuk effect in ultrahigh magnetic fields,” is misleading.
2. According to the authors, the metal-insulator transition for $x=1$ sample is accompanied by a structural phase transition. While here, they want to claim a purely electronic Pomeranchuk effect. This needs to be more clearly addressed.
3. What is the magnetic response for the $x = 0.45$ salt, which shows the re-entrant metallic state at low temperatures?

Other minor suggestions and points needed to be addressed:

1. No page number is labeled.
2. Page 3, “Only a few examples of electrons exhibiting the Pomeranchuk like effect in solids have been reported”. In recent years, the Pomeranchuk-like effect has been observed in more 2D systems, including TMDCs moiré systems (*Nature* 597, 350-354 (2021), *Phys. Rev. X* 12, 041015 (2022),) and rhombohedral multilayer graphene system (*Nature* 640, 355–360 (2025)).
3. Page 4, “Chemical pressure, the counterpart of physical pressure in ^3He , was introduced by mixing deuterated DMe-DCNQi (d8-DMe-DCNQi)”, How does the chemical pressure P depend on the d8-DMe DCNQi fraction x ?
4. In Page 6, “In the lower-temperature region of the $x = 0.3$ salt, the slope of the phase boundary exhibits $dH/dT < 0$, coinciding with the emergence of the reentrant metallic state.”

This is a bit confusion, coinciding with the reentrant metallic state in $x=0.45$ salt?

5. In Figure 2a, why is the ρ - T curve not smooth for $x = 0.45$?
6. In Figure 2b, 2c, and 2d, what do the light-colored curves represent? It does not seem to have been mentioned anywhere in the manuscript. It's likely the down-ramped curve. Would the hysteresis cause by temperature difference? Why aren't the lighter-colored curves shown for the two middle panels of Figure 2c?
7. Page 4, "These results indicate that the region of the insulating phase in the pressure temperature phase diagram expands toward the lower-pressure side under magnetic fields", authors could draw a schematic diagram for this statement.
8. Page 5, "Figure 2d illustrates the relative phase change $\Delta\phi$ of RF waves as a function of the magnetic field at 100 K and 110 K", the temperatures labeled in Figure 2d are 98 K and 107 K, respectively.
9. In Figure 3a, the authors should state how the data uncertainties were determined.
10. The authors interpret the noise in Fig. 2f and 2g as the damage to the sample caused by eddy currents. Is there further evidence of sample damage? Is it feasible to reduce dH/dt ? Do the authors have the data for decreasing magnetic field conditions in the EMFC system?